# Development of Antioxidant-Loaded Nanoliposomes Employing Lecithins with Different Purity Grades

**DOI:** 10.3390/molecules25225344

**Published:** 2020-11-16

**Authors:** Cristhian J. Yarce, Maria J. Alhajj, Julieth D. Sanchez, Jose Oñate-Garzón, Constain H. Salamanca

**Affiliations:** 1Laboratorio de Diseño y Formulación de Productos Químicos y Derivados, Departamento de Ciencias Farmacéuticas, Facultad de Ciencias Naturales, Universidad ICESI, Calle 18 No. 122-135, 760035 Cali, Colombia; cjyarce@icesi.edu.co (C.J.Y.); mariajoalhajj@hotmail.com (M.J.A.); julieths28@outlook.com (J.D.S.); 2Centro de Ingredientes Naturales Especializados y Biotecnológicos-CINEB, Facultad de Ciencias Naturales, Universidad ICESI, Calle 18 No. 122-135, 760035 Cali, Colombia; 3Facultad de Ciencias Básicas, Programa de Microbiología, Universidad Santiago de Cali, Calle 5 No. 62-00, 760035 Cali, Colombia; jose.onate00@usc.edu.co

**Keywords:** antioxidant effect, lectins, nanoliposomes, purity grade, quercetin, trans-aconitic acid

## Abstract

This work focused on comparing the ability of lecithins with two purity grades regarding their performance in the development of nanoliposomes, as well as their ability to contain and release polar (trans-aconitic acid) and non-polar (quercetin) antioxidant compounds. First, the chemical characterization of both lecithins was carried out through infrared spectroscopy (FTIR), electrospray ionization mass spectrometry (ESI/MS), and modulated differential scanning calorimetry (mDSC). Second, nanoliposomes were prepared by the ethanol injection method and characterized by means of particle size, polydispersity, and zeta potential measurements. Third, the encapsulation efficiency and in vitro release profiles of antioxidants were evaluated. Finally, the antioxidant effect of quercetin and trans aconitic acid in the presence and absence of nanoliposomes was assessed through the oxygen radical absorbance capacity (ORAC) assay. The results showed that, although there are differences in the chemical composition between the two lecithins, these allow the development of nanoliposomes with very similar physicochemical features. Likewise, nanoliposomes elaborated with low purity grade lecithins favored the encapsulation and release of trans-aconitic acid (TAA), while the nanoliposomes made with high purity lecithins favored the encapsulation of quercetin (QCT) and modified its release. Regarding the antioxidant effect, the vehiculization of TAA and QCT in nanoliposomes led to an increase in the antioxidant capability, where QCT showed a sustained effect over time and TAA exhibited a rapidly decaying effect. Likewise, liposomal systems were also found to have a slight antioxidant effect.

## 1. Introduction

Lecithins are a type of raw material widely used by different sectors, such as pharmaceutical, food, and cosmetic sectors, where they are employed for different purposes due to their great versatility [1]. Some of these uses correspond to their ability to act as emulsifiers, as well as humectants and suspension stabilizers [2]. However, their most important application lies in their ability to form nanoliposomes, which represent a very special type of self-assembling system that is very useful as a nanometric formulation vehicle for multiple compounds of interest in the previously mentioned sectors [3,4].

Lecithins are complex raw materials because they consist of a mixture of various types of phospholipids, where phosphatidylcholine (PC) is usually the main component [5]. However, they may also contain other types of phospholipids, such as phosphatidylethanolamines (PE), phosphatidylserines (PS), phosphatidylinositols (PI), phosphatidylglycerols (PG), and glyphospholipids (GLP), as well as other components, such as phosphatidic acid (PA) and cholesterol [6]. Therefore, depending on the obtention source (vegetable or animal), the type of extraction, and the purification methods, lecithins can be composed of different types and amounts of phospholipids and thus have a wide diversity of references with multiple purity grades and commercial prizes [7]. This diversity of references can, in some cases, lead to confusion or a lack of criteria for choosing the correct lecithin. In the case of the pharmaceutical sector, this choice is easier to make due to the intrinsic features of such products and their regulatory affairs, in which it is practically mandatory to use raw materials with the highest possible purity grade. In contrast, this situation is different in other sectors, such as foodstuff and cosmetic sectors, where lower purity grade lecithins can be used with less restrictions. Therefore, the question presented in this study is as follows: Can a low purity lecithin develop nanoliposomes, just like a high purity lecithin? To address this question, the work focused on (i) chemically characterizing two types of soybean lectins with different purity grades and commercial value; (ii) the development and characterization of nanoliposomes loaded with two model antioxidants, corresponding to quercetin [8] and trans-aconitic acid [9]; and (iii) determining the encapsulation efficiency and in vitro release profiles of antioxidant compounds from nanoliposomal vehicles.

## 2. Results and Discussion

### 2.1. Lecithin Characterization

Firstly, it is important to highlight that for the comparison of the purity grade to be consistent, it is necessary that lecithins come from the same origin and therefore have a similar phospholipid composition. If this was not the case, the results would be meaningless and may not be reproduced. Considering this, the results discussed as the conclusions issued in this manuscript correspond to lecithins from a similar plant source (soybean). In the case of low purity lecithin, this was found to be a brown fluid with a sticky consistency. In contrast, the high purity lecithin was shown to be a yellowish granular waxy solid that could be easily handled. Such physical appearance characteristics are a very important factor to consider in foodstuff raw materials, since, depending on their consistency, various stages of product development, such as dispensing, weighing, and post-production cleaning, may be easier to carry out. Therefore, low purity lecithin has a disadvantage compared to high purity lecithin, which is easier to handle. Additionally, the color of the raw materials is another aspect to be considered, since the appearance of these plays an important role in the sensory characteristics of the product to be developed, where dark colors are more difficult to mask, which may be another disadvantage of low purity lecithin versus high purity lecithin. Consequently, the physical appearance of these lecithins is largely determined by their own chemical composition. In this way, and according to the respective technical data sheets, low purity lecithin has a distribution of phospholipids of around 90%, where ~50% is phosphatidylcholine, ~30% are inositol phosphatides, and ~10% is phosphatidylethanolamine; the remaining 10% corresponds to several impurities [5,10]. In contrast, high purity lecithin consists of 97% phospholipids, where ~92% is phosphatidylcholine, ~3% is lysophosphatidylcholine, and ~2% are other phosphatides, with the remainder also corresponding to compounds other than phospholipids [5].

#### 2.1.1. FTIR Characterization

Figure 1A shows the FTIR spectra for low and high purity lecithins and also indicates the chemical structures of the two phospholipids that, according to the literature, are present in a greater proportion in lecithins [5]. In the case of Lp-SBL, signals were found at 3384, 3010, 2924, and 2854 cm^−1^. These signals correspond to the stretching vibrations of the OH groups from (i) the carboxylic acid function of the fatty acids and the saccharides present in glyphospholipids; (ii) the amino group present in some phospholipids, such as phosphatidylcholine; and (iii) the symmetric and asymmetric tension vibrations of the CH_2_ groups present in fatty acids. In the case of Hp-SBL, the signals were observed at 3334, 3009, 2924, and 2854 cm^−1^. Moreover, the OH band amplitude and intensity suggest that low purity lecithin has a higher content of saccharides. On the other hand, the bands of 1744 cm^−1^ for Lp-SBL and 1739 cm^−1^ for Hp-SBL correspond to the stretching vibration of the carbonyl bond (C=O). However, the signal shifting suggests that the substitution for carbonyl is different in each raw material, which is due to differences in the length and proportion of substituent fatty acids in phospholipids. Likewise, the multiple bands that appear for both raw materials between 1238 and 1064 cm^−1^ correspond to the phosphate groups in the phospholipid chains. Finally, the wide shapes of the signals presented between 1064 and 1165 cm^−1^ in low purity lecithin are representative of the asymmetric tensions of OH groups present in sugars, which is consistent with a higher amount of saccharide molecules in that lecithin.

#### 2.1.2. Thermal Analysis

The results of the thermal characterization performed are shown in Figure 1B. In the case of low purity lecithin, it is possible to observe two thermal transitions. The first one had an onset at approximately 38.2 °C, a transition peak at around 91.6 °C, and an enthalpy value of 132.3 J/g. This behavior can be observed in the total heat flow signal (solid line) and in the non-reversible heat flow signal (short dashed line), but not in the reversible heat flow signal (dashed line). This indicates that there is an endothermic-type transition caused by the kinetics of the molecules inside the material [11,12,13]. It leads to a reorganization and presents multiple rearrangements in their physical structure, suggesting a possible loss of volatile substances and water present in the Lp-SBL raw material. Consequently, according to the enthalpy of the transition, the energy expenditure required to remove volatile substances and water was relatively high, due to the contribution of the heat of water vaporization [12,13,14,15,16,17]. The DSC results for Lp-SBL are consistent with the mass spectra, where representative signs of mass loss due to the dehydration of saccharide-like molecules were observed.

On the other hand, in the case of high purity lecithin, the total heat flow signal presents an endothermic-type transition, with an onset at around 164.4 °C, a peak at 181.8 °C, and an enthalpy of 6.5 J/g. These signals are consistent for the reversible heat flux and the non-reversible heat flux. This shows that the high purity lecithin is a consistent material and that its composition is defined by compounds that operate around approximate temperature values. Furthermore, due to the relatively low enthalpy of the transition, it establishes the heat flow through the material being transmitted homogeneously, which requires a low energy expenditure for the fusion of phospholipids in Hp-SBL. Furthermore, for this raw material, there is no sign of water loss or signs due to the rearrangement of the material, as is the case for Lp-SBL. Finally, all these results are reliable in terms of the fluidity and physical appearance of the Lp-SBL, where it is necessary to remove the water and other volatile components prior to the fusion of the phospholipids, which is a transition that is displaced with an onset of 211.0 °C, with a peak at 220.2 °C and an enthalpy of 1.96 J/g.

#### 2.1.3. Electrospray Ionization Mass Spectrometry (MS/ES^−^/ES^+^)

Figure 2 shows the mass spectra recorded in negative (ESI^−^) and positive (ESI^+^) ion mode. The signals are discussed together for both Lp-SBL and Hp-SBL. In the case of the mass spectrum in ESI^−^ negative ionization mode, (Figure 2A left), the [M + H] ion is observed for phosphatidylcholine substituted with linoleic acid (C18:2) and palmitic acid (C16:0), at an *m*/*z* ratio of 758 Da. Additionally, an ion with an *m*/*z* of 833 Da, corresponding to phosphatidyl inositol, with substitutions of linoleic acid and oleic acid (C18:1), is observed. Conversely, in the case of the mass spectrum registered in positive ESI^+^ mode (Figure 2A right), the presence of a molecular adduct of potassium for phosphatidylcholine, which is the ion [M + K] with an *m*/*z* ratio, can be found at 797 Da. In addition, the ion corresponding to the dehydrated form of [GLP] can be found at an *m*/*z* ratio of 832 Da. It is important to highlight that, in the case of low purity lecithin, the abundance between [M + H] and [GLP] is inverted regarding the abundance of these ions in high purity lecithin. Consequently, a greater abundance is observed for the phosphatidyl inositol signal than for the phosphatidylcholine signal, which is consistent with a higher presence of saccharides in the low purity lecithin.

On the other hand, Figure 2B shows the estimated fragmentation patterns in negative ionization mode for phosphatidylcholine and inositol glycophospholipid, respectively. It can be observed that the molecular ion [M + H] for phosphatidylcholine suffers a loss of fraction (1) of *m*/*z* 279 Da, which corresponds to the loss of the linoleic acid substituent of phosphatidylcholine, which leads to the generation of 479 Da m/fragment of z (3). In addition, a loss of *m*/*z* of 255 Da, corresponding to the palmitic acid substituent and a fragment (4) of the *m*/*z* ratio of 714 Da, is formed from the rupture of the nitrogen-bound methyl groups of the choline fraction. In Figure 3B, a water molecule from the phosphatidyl inositol [GLP] glycophospholipid is lost and forms a fragment with an *m*/*z* ratio of 831 Da. These processes occur in both low purity lecithin and high purity lecithin. Furthermore, Figure 3C shows the fragmentation patterns for phosphatidylcholine and inositol glyphospholipids in positive ionization mode. In ESI^+^, a molecular adduct for phosphatidylcholine [M + K] is presented, which suffers losses of the substituent fatty acids in the phospholipid (1) and the methyl groups bounded to nitrogen in the choline fraction (4). In the case of [GLP], a loss of the dehydrated inositol molecule (5) and the substituent fatty acids in the phospholipid (2) can be observed. Accordingly, it is important to highlight that the results suggest that the ions present in the low purity lecithin are also present in high purity lecithin. Therefore, phospholipids such as phosphatidylcholine and phosphatidyl inositol are present in both raw materials and it is thus possible to obtain the advantages of these compounds when using lecithins for a process of transformation or the subsequent preparation of a food or cosmetic product. However, it is estimated that the phosphatidylcholine:phosphatidyl inositol ratio is lower for Lp-SBL compared to Hp-SBL. Furthermore, it should be recognized that other phospholipids, such as phosphatidylethanolamine and phosphatidylserine, may also be present in the raw materials [10], but their signals were not observed, because they could be overlapped with the abundance of ions from other compounds in lecithins.

### 2.2. Development of Nanoliposomes

#### 2.2.1. Physicochemical Characterization 

Figure 3 shows the results of the particle size, polydispersity index, and zeta potential for nanoliposomes prepared with low purity (Lp-SBL) and high purity (Hp-SBL) soybean lecithins. In the case of nanoliposomes prepared with Lp-SBL, the particle size of non-loaded nanoliposomes (BLK-Lp) was around 150 nm, while the particle sizes of the nanoliposomes loaded with QCT (QCT-Lp) and TAA (TAA-Lp) were ~250 and ~200 nm, respectively. In contrast, for the nanoliposomes prepared with Hp-SBL, the size of non-loaded nanoliposomes (BLK-Hp) was around 200 nm, while the sizes of the nanoliposomes loaded with QCT (QCT-Hp) and TAA (TAA-Hp) were ~250 and ~180 nm, respectively (Figure 3A). These differences in particle size can be explained considering the composition of phospholipids in both lectins and specifically, in those that consist of greater polar phospholipid amounts (inositol, saccharides, etc.), such as Lp-SBL, and that lead to a higher compaction of the aqueous liposomal core. Likewise, such a compaction effect can also be observed when the nanoliposomes are loaded with TAA, which is a very polar molecule [18,19,20,21,22]. On the other hand, it is important to highlight that the particle size reached in the liposomes prepared with both lecithins is on a nanometric scale (150–250 nm), which favors the permeation process in many biological membranes and is, in fact, an interesting characteristic for the development of functional food products [23,24,25]. Regarding the polydispersity values (Figure 3B), it was found that, regardless of the lecithin used, it always tends to form nanoliposomes (loaded and unloaded) with a low particle size distribution (PDI < 0.3). This result is very interesting, since it suggests that both lecithins can be used to produce nanoliposomes of a regular size, which is essential to guaranteeing the uniformity of the content inside the nanoliposomes [4,26,27]. In relation to the zeta potential (Figure 3C), this parameter showed values of around −30 mV in all of the nanosystems obtained, regardless of the lecithin used. These negative zeta potential values can be explained considering that liposomes consist of a small portion of fatty acids (oleic acid, linoleic acid, etc.), which can be slightly ionized in the aqueous medium. In the same way, it can be considered that the autoprotolysis effect of water, where some hydroxy anions are generated and located in the liposome-aqueous medium interface, leads to an increase in the negative zeta potential [28]. Therefore, this result suggests that nanoliposomes could have an adequate physical stability, because of electrostatic repulsion that prevent interparticle aggregation [29,30,31]. Furthermore, the systems loaded with TAA lead to more negative zeta potential values. This result can be explained by considering that TAA is a polycarboxylic acid (carboxylic acid-carboxylate), which can also be adsorbed in the interfacial zone (a fraction), increasing the anionic charge in this zone and therefore, the zeta potential. 

#### 2.2.2. Encapsulation and In Vitro Release of Antioxidant Compounds

Figure 4A shows the results of the encapsulation efficiency (EE) of antioxidant compounds (QCT and TAA) from nanoliposomes prepared with low purity (Lp-SBL) and high purity (Hp-SBL) soybean lecithins. In the case of nanoliposomes loaded with QCT and TAA and prepared with Lp-SBL, the results of the encapsulation efficiency were around 88% and 57%, respectively. For the nanoliposomes loaded with QCT and TAA and prepared with Hp-SBL, the results of the encapsulation efficiency were around 99% and 27%, respectively. These results suggest that the Lp-SBL favored the encapsulation of polar compounds, while the Hp-SBL favored non-polar compound encapsulation. This result may be explained considering the differences between the lecithins’ composition, where Lp-SBL displayed a higher amount of glycophospholipids and sugars, which could interact with polar compounds such as TAA through hydrogen bond interactions. Another study also suggested that some small molecules, such as caffeine, are mainly located in the solvation layer adjacent to the liposomal lipid bilayer interface [32]. On the contrary, the Hp-SBL exhibited a composition of phospholipids with a non-polar character and where the encapsulation of QCT (non-polar) could possibly take place by a process such as micellar solubilization.

On the other hand, Figure 4B shows the in vitro release of antioxidant compounds (QCT and TAA) from nanoliposomes prepared with low purity (Lp-SBL) and high purity (Hp-SBL) soybean lecithins. In the case of QCT, it is noteworthy that the release profiles display a lag time of 240 min for nanoliposomes made with Lp-SBL and a maximum released amount of 15%. In comparison, for nanoliposomes elaborated with Hp-SBL, the lag time was 60 min, and the maximum released amount was 40%. These results are very interesting, since they show that the nanoliposomal vehicle considerably affects the QCT release, which can be easily appreciated when these are compared against the control (QCT alone, i.e., without a nanoliposomal vehicle), which is faster and exhibits a greater amount (99%). This result is very consistent, considering that the QCT is encapsulated within the hydrophobic pseudo-phase formed by the lamellar structures of the hydrocarbon chains of phospholipids. Likewise, this result supports the previous results of the encapsulation efficiency, where the QCT is encapsulated in high amounts in both nanoliposomal systems (88–99%), describing a greater affinity for the vehicle than the aqueous medium and therefore, its release is slow and controlled, as reflected in their respective lag times. On the contrary, the TAA-loaded nanoliposomes showed a faster release profile, regardless of the type of lecithin used. In this way, the TAA release from nanoliposomes described a similar behavior to that shown by the control (TAA alone), where rapid diffusion was observed. Regarding the control, it could be seen that around 80% of the TAA went through the bi-compartmental system in the first 5 min, and then remained almost constant, suggesting that the material balance had been reached. In the case of the TAA loaded in the nanoliposomes, it was found that the release was lower (~60%), which indicates that most of the TAA is in the interfacial zone and not within the liposomal aqueous core and therefore, the release is practically immediate. Likewise, the difference in the amount of TAA between the control and the nanoliposomal systems suggests that there is a fraction of TAA inside the nanoliposome and to achieve a higher released amount, a longer time or other external conditions would possibly be required. Similar results have also been reported, where it has been described that, depending on the molecule polarity, the encapsulation and release mechanisms may vary [32,33,34,35,36,37].

### 2.3. Antioxidant Effect Assay

The ORAC assay results for pure QCT and TAA, as well as loaded in nanoliposomes, are shown in Figure 5. In the case of Trolox^®^ (standard), QCT, and TAA antioxidants, the increase in their concentrations prevented fluorescence quenching of the probe in different ways. In the case of pure QCT, a similar trend to that described by the standard antioxidant was observed, while TAA exhibited a different behavior. The maximum antioxidant effect was reached at concentrations of 30.5, 40.5, and 140 µg/mL for Trolox^®^, QCT, and TAA, respectively (Figure 5A). These results can be explained by considering the photo-physical mechanism that takes place in the ORAC assay, as well as analyzing the chemical structures of antioxidants (Figure 5B). In this context, fluorescein in aqueous medium emits fluorescent radiation at 520 nm, which remains practically unchanged over time. Then, the addition of the AAPH reagent leads to the oxidation of fluorescein by the reactive oxygen species (ROS) generated during the homolytic cleavage of such reagent [38,39]. Consequently, the oxidation in fluorescein leads to a decay of the fluorescence intensity over time (quenching) [38,39,40]. Therefore, the addition of an antioxidant establishes competition with fluorescein in the oxidation process and thus, when the antioxidant interacts with the ROS species, fluorescence decay is avoided, which is interpreted as an antioxidant effect. In these molecules, the phenyl substituent presents a thermodynamic equilibrium between phenyl and phenolate species, where the phenolate form is the one that reacts with the ROS species [8,38,39,41,42]. On the contrary, the antioxidant effect of TAA is due to the alteration of the phenyl-phenolate thermodynamic balance in fluorescein, leading to a slight predominance of the neutral form (R-phenyl), which is less reactive against ROS species. Moreover, the antioxidant effect was achieved at high concentrations (140 µg/mL), being moderate. However, it is important to note that TAA does not have oxidizable groups [9,43], whereas the Trolox^®^ and QCT antioxidants do and therefore, TAA’s ability to avoid fluorescence decay is more limited.

Regarding nanoliposomes loaded with QCT and TAA (Figure 5C), the antioxidant effect changes, depending on the type of compound. In the case of pure QCT, slight fluorescence decay of the probe was observed after 30 min, while QCT-loaded nanoliposomes displayed a sustained fluorescence effect. This result is consistent with that previously obtained in the QCT release profile from liposomes, where a sustained release was observed. Therefore, the use of nanoliposomes extends the antioxidant effect of QCT over time. On the contrary, TAA presented a considerable change when it was loaded in nanoliposomes, leading to a remarkable recovery of fluorescence. In the case of pure TAA, the antioxidant effect was very short and after 30 min, the probe fluorescence was considerably quenched. However, when TAA was loaded on nanoliposomes, there was a slight recovery of the fluorescent emission, suggesting a slight increase in the antioxidant effect. However, this recovery was less than that obtained with Trolox^®^ and QCT antioxidants. This result is consistent considering that TAA does not have the phenyl substituent, which is involved in the oxidation by ROS species.

Regarding the antioxidant efficiency (Figure 5D), the results showed that pure QCT and that loaded in the nanoliposomes have practically the same antioxidant efficiency of around 70%, being very similar to that shown by the standard antioxidant. Nevertheless, the most interesting result is the sustained antioxidant effect over time, which was achieved when QCT was loaded inside the liposomes. In relation to TAA, the antioxidant efficiency was <30%, which is lower than the values for Trolox^®^ and QCT antioxidants. However, this antioxidant efficiency was significantly improved when TAA was loaded in the nanoliposomes, resulting in an antioxidant efficiency value of around 60%. Regarding the type of lecithins used for the formation of nanoliposomes, no significant changes were observed concerning the antioxidant effect. In the case of Lp-SBL-QCT and Hp-SBL-QCT systems, as well as Lp-SBL-TAA and Hp-SBL-TAA systems, the antioxidant efficiencies were very similar to each other, with values of around 70% and 60%, respectively. Likewise, it was observed that both liposomal systems also exhibit a slight antioxidant effect of around 20%. These results are very interesting because they show that nanoliposomes formed with low purity lecithins can provide a similar antioxidant effect to that provided with nanoliposomes formed with high purity lecithins.

## 3. Material and Methods

### 3.1. Materials

Low purity grade lecithin (Lp-SBL) was obtained from Farmacia-Drogueria San Jorge Ltd.a (Cali, Colombia), whereas high purity grade lecithin (Hp-SBL) (Epikuron 200™, Mw = 786 g/mol) was purchased from Cargill Corporation (Wayzata, MN, USA). Both lectins claim to come from the same plant source (soybean) and were used as received. The phospholipid 1,2-dioleoyl-sn-glycero-3-phosphoethanolamine (DOPE, Mw = 744.03 g/mol) and cholesterol (Mw = 386 g/mol) were purchased from Avanti Polar Lipids (Alabaster, AL, USA). Quercetin (QCT), trans-aconitic acid (TAA), and Ethanol USP grade were purchased from Sigma-Aldrich (St. Louis, MO, USA) and ultrapure water was supplied by an Elix Essential Millipore^®^ purification system, with a mean conductivity value of ~0.050 µScm. Methanol Lichrosolv™ mass spectrometry grade was obtained from Sigma-Aldrich (St. Louis, MO, USA). Regarding the antioxidant activity assay, the analytical reagents employed were fluorescein sodium salt, 2,2′-Azobis(2-methylpropionamidine) dihydrochloride (AAPH), 6-Hydroxy-2,5,7,8-tetramethylchromane-2-carboxylic acid (Trolox), potassium phosphate monobasic, and potassium phosphate dibasic, which were purchased from Sigma-Aldrich (Merck KGaA, Darmstadt, Germany)

### 3.2. Chemical Characterization of Lecithins 

#### 3.2.1. Infrared Spectroscopy (FTIR) Characterization

The structural characterization of lecithins was performed in an FT-infrared spectrometer coupled to an attenuated total reflectance (ATR) instrument (Nicolet 6700, Thermo Fisher Scientific, Waltham, MA, USA). The spectra were recorded by using an attenuated total reflectance Smart iTR™ accessory, where the spectra of both lectins were compared.

#### 3.2.2. Thermal Analysis

Thermal studies were carried out on a Q2000 differential scanning calorimeter (DSC; TA Instruments, New Castle, DE, USA) calibrated with indium T_m_ = 155.78 °C and ∆H_m_ = 28.71 J/g. Therefore, a modulated heating–cooling cycle from −10 °C (263.15 K) to 250 °C (523.15 K) at a heating rate of 5 °C/min was used. It was applied a ±0.5 °C modulation each 40 s. Approximately 10 mg of each sample was placed on a hermetic crucible with a lid, and an empty hermetic crucible was used as a reference.

#### 3.2.3. Electrospray Ionization Mass Spectrometry

A methanolic solution of each respective lecithin was prepared at a concentration of 1 mg/mL, which was injected by direct infusion (flow of 3 µL/min) into a simple quadrupole mass spectrometer coupled to an electrospray ionization source (SQD2/ESI, Waters Corporation, Milford, MA, USA). The spectra were obtained in total ion scanning mode for positive ions (ESI^+^) and negative ions (ESI^−^), in a range of 200 to 2000 *m*/*z*. The test conditions for both ionization modes were as follows: Desolvation gas flow, 550 L/h; desolvation temperature, 500 °C; source temperature, 150 °C; extraction voltage, 3 V; cone voltage, 40 V; and capillary voltage 2.69 kV. The samples were injected in triplicate.

### 3.3. Development of Nanoliposomes

#### 3.3.1. Preparation by the Ethanol Injection Method

The nanoliposomes were prepared based on a sequential process defined in several steps, depending on the antioxidant [44]. In the case of QCT-loaded nanoliposomes, they were prepared as follows: (i) Dispersion of phospholipids in organic phase: Ethanolic solutions of lecithin (1.3 mg/mL), cholesterol (0.64 mg/mL), DOPE (1.23 mg/mL), and QCT (350.8 µg/mL) were prepared. From those solutions, volumes of 30, 11.5, 30, and 28.5 μL were taken, respectively; (ii) mixture with aqueous phase: 100 μL of organic phase was slowly added to 100 μL of ultrapure water, and the solution was then stirred for 1 min and left to rest for 10 min; and (iii) Nanoliposome formation: The resulting mixture between the organic phase and aqueous media was diluted in 800 μL of ultra-pure water. This process led to nanoliposome formation with a final QCT concentration of 35 µg/mL. In the case of TTA-loaded nanoliposomes, these were prepared as follows: (i) Dispersion of phospholipids in organic phase: Ethanolic solutions of lecithin (1.3 mg/mL), cholesterol (0.64 mg/mL), and DOPE (1.23 mg/mL) were prepared. From those solutions, volumes of 42.4, 15.2, and 42.4 μL were taken, respectively. (ii) Mixture with aqueous phase: 100 μL of organic phase was slowly added to 100 μL of aqueous media trans-aconitic acid solution with a 1000 µg/mL concentration (Ultra-pure water), and the sample was then stirred for 1 min and left to rest for 10 min. The nanoliposome formation was conducted similarly to QCT-loaded nanoliposomes, where the TAA concentration in nanoliposomes was 100 µg/mL. Liposomes were purified by means of the ultrafiltration/centrifugation technique. An aliquot of each nanoliposome dispersion was transferred into an ultrafiltration tube (VWR, modified polyethersulfone-PES 10 kDa, 500 µL, diameter: 0.96 cm) and centrifuged (MIKRO 185, Hettich Lab Technology, Tuttlingen, Germany) at 10,000 rpm (1075 RFC) for 6 min. Each nanoliposomal system was prepared in triplicate at room temperature (25 °C).

#### 3.3.2. Physicochemical Characterization 

The particle size and zeta potential were determined using a Zetasizer nano ZSP (Malvern Instrument, Worcestershire, UK) with a red He/Ne laser (633 nm). The particle size was measured using dynamic light scattering (DLS) with an angle of scattering of 173° at 25 °C, in a quartz flow cell (ZEN0023), whereas the zeta potential was measured using a disposable folded capillary cell (DTS1070). This instrument reports the particle size as the mean particle diameter (z-average), with the polydispersity index (PDI) ranging from 0 (monodisperse) to 1 (very broad distribution). All measurements were performed in triplicate after an appropriate dilution (~5:5000, *v*/*v*) of the liposome suspension in ultra-pure water and are reported as the mean and standard deviation of measurements made from freshly prepared liposomal dispersions.

### 3.4. Encapsulation and In Vitro Release of Antioxidant Compounds

#### 3.4.1. Antioxidant Encapsulation Efficiency (EE)

The EE of QCT and TAA was assessed using the ultrafiltration/centrifugation technique. An aliquot of each nanoliposome dispersion was transferred into an ultrafiltration tube (VWR, modified polyethersulfone-PES 10 kDa, 500 µL, diameter: 0.96 cm) and centrifuged (MIKRO 185, Hettich Lab Technology, Tuttlingen, Germany) at 10,000 rpm (1075 RFC) for 6 min. For QCT quantification, an aliquot of the filtrate obtained in each system was obtained and evaluated with a microplate reader (Synergy, H1. Microplate reader, Biotek, Winooski, VT, USA). The amount of quercetin was determined by interpolation from a calibration curve that was previously prepared at the following concentrations, using ultra-pure water as the solvent: 1, 3, 10, 30, and 300 µg/mL. For the quantification of TAA, an aliquot of the filtrate obtained in each system was obtained and evaluated via HPLC (Lachrom elite, Merck, Darmstadt, Germany) equipped with a photo diode array (PDA) detector and an automatic sampling system. The mobile phase consisted of acetonitrile and water with a pH of 2.5 (10:90), and the flow rate was 0.8 mL/min. Separation was achieved using a 50 mm × 4.6 mm, Zorbax Eclipse XDB-C18 (Agilent technologies, Santa Clara, CA, USA) reversed-phase column, with an average particle size of 1.8 µm, keeping the column at 25 °C. The column effluent was monitored at 270 nm, and the chromatographic data analysis was performed with EZChrome software (Agilent technologies, Santa Clara, CA, USA). The amount of TAA was determined by interpolation from a calibration curve that was previously prepared at the following concentrations, using ultra-pure water as the solvent: 5, 10, 20, 40, 80, and 100 µg/mL. Finally, the amount of QCT and TAA loaded inside the nanoliposomes was calculated using the following equation:(1)$$EE=100-\left[\frac{Q_{t}-Q_{s}}{Q_{t}}\times 100\right]$$
where *EE*, *Q_t_*, and *Qs* correspond to the encapsulation efficiency, initial total amount of bioactive compound, and amount of bioactive compound in the filtrate, respectively.

#### 3.4.2. In Vitro Antioxidant Release

The in vitro release of QCT and TAA from nanoliposomes was assessed by the dialysis method, using 5 mL of phosphate buffer with a pH of 7.0 and 150 mM aqueous medium under sink conditions. Therefore, volumes of 500 µL of the nanoliposomes were placed into a dialysis tube (VWR, modified polyethersulfone-PES cut-off 10 kDa, 500 µL, diameter: 0.96 cm) in triplicate and dialyzed at 37 °C for 16 h with constant stirring in an incubated orbital shaker (Inkubator 1000 with Unimax 1010, Heidolph Instruments, Schwalbach, Germany). Subsequently, the samples were taken from the external medium at intervals of 0, 5, 10, 20, 30, 60, 120, 180, 240, 360, and 996 min. The quantification of QCT and TAA was performed as described in the encapsulation efficiency section.

### 3.5. Antioxidant Effect Assay

The antioxidant activity was determined by the oxygen radical absorbance capacity (ORAC) assay [45,46], which is a method based on an evaluation of the ability of a compound to prevent the fluorescence quenching mediated by the AAPH reagent. For this, fluorescein (a fluorescent probe) and AAPH solutions were prepared in PBS (pH: 7.0) at concentrations of 0.02 mg/mL and 59.8 mg/mL, respectively. In contrast, the antioxidant compounds were made in PBS (pH: 7.0) at concentrations of 7.65, 15.25, 30.5, and 61 µg/mL for the standard antioxidant (Trolox^®^); 5, 10, 20, 40, 80, and 100 µg/mL for QCT; and 20, 40, 80, 100, 120, and 140 µg/mL for TAA. The evaluation of the fluorescent decay for fluorescein was conducted using a Synergy H1 microplate reader (Biotek, Winooski, VT, USA), where excitation and emission wavelengths of 485 and 520 nm were used, respectively, at 37 °C. Measurements were carried out in triplicate every 3 min for 2 h, and the data obtained from the fluorescent vs. time curves are reported as the average antioxidant efficiency (AE) of the antioxidant compound. This parameter is defined as the area under the fluorescent decay curve (AUC) recorded at a particular time in relation to the rectangular area (R) described by 100% of fluorescent emission of pure fluorescein (AUC of the positive control) at the same time. Therefore, the antioxidant efficiency can be calculated from:(2)$$AE=\frac{AUC}{R}\times 100\%$$

Finally, the pure fluorescein solution was named as the positive control because fluorescence decay does not take place. On the contrary, the fluorescein + AAPH solution was labeled as the negative control, because AAPH forms reactive oxygen species (ROS) that lead to a high quenching of the fluorescent probe. Similarly, Trolox^®^ and unloaded nanoliposomes were labeled as the standard antioxidant and blank, respectively. 

### 3.6. Statistical Analysis

The data were tabulated and analyzed using Microsoft Excel and Graph Pad Prism, respectively. The homogeneity of variance in the data was analyzed using Bartlett’s test. Statistical comparisons were conducted using a one-way ANOVA. The Bonferroni post-hoc test was used to determine significant differences between the two independent groups. A confidence level of 95% was adopted. Data are expressed as the mean ± standard deviation.

## 4. Conclusions

In this study, it was established that low and high purity lecithins from a similar plant source (soybean) show differences and similarities, which can mean both advantages and disadvantages for their use as raw materials. First, the physical appearance is a determining factor for obtaining adequate handling and organoleptic characteristics at different stages of the product life cycle. Accordingly, the physical handling of a material such as low purity degree lecithin presents some difficulties compared to a material such as high purity degree lecithin. However, the physicochemical characterization of both materials using instrumental techniques such as FTIR, MS, and DSC, indicated that the chemical compositions of the two lecithins are very similar. Therefore, phospholipids such as phosphatidylcholine and phosphatidyl inositol were the main constituents of lecithins in this work. Regarding the ability to form nanoliposomal systems with adequate physicochemical characteristics (particle sizes < 300 nm, PDI < 0.3 nm, and zeta potential values of ~−30 mV), it was found that both lectins allow the preparation of these types of soft nanometric vehicles in a simple way. On the other hand, it was established that lecithins with a low purity grade consist of a greater amount of polar phospholipids, which tend to mainly encapsulate trans-aconitic acid (TAA). In contrast, lecithins with a high purity degree and which mainly consist of non-polar phospholipids tend to encapsulate a higher amount of quercetin (QCT). It was also established that the release of antioxidant compounds from nanoliposomal systems depends on their polarity and the way that they are encapsulated. In the case of the TAA, this is mainly located in the nanoliposome-water interface, where its release is very fast and around 60%. On the contrary, QCT release is slow and occurs at smaller quantities (15–40%), which is explained by the specific location of QCT within the lamellar structure of the nanoliposomes, where its affinity for such pseudo phase is great and therefore, its release is limited. It was demonstrated that low purity lecithins represent a viable alternative in terms of costs–benefits for obtaining innovative products for application to the food and cosmetic sectors. On the other hand, it was found that QCT presents a high antioxidant efficiency, which is sustained over time, describing a behavior very similar to the standard. On the contrary, it was found that TAA has a low antioxidant efficiency that can be increased when it is loaded with nanoliposomes. Finally, it was found that nanoliposomes formed with low purity lecithins can provide an antioxidant effect equal to that provided with nanoliposomes formed with high purity lecithins.

## Figures and Tables

**Figure 1 molecules-25-05344-f001:**
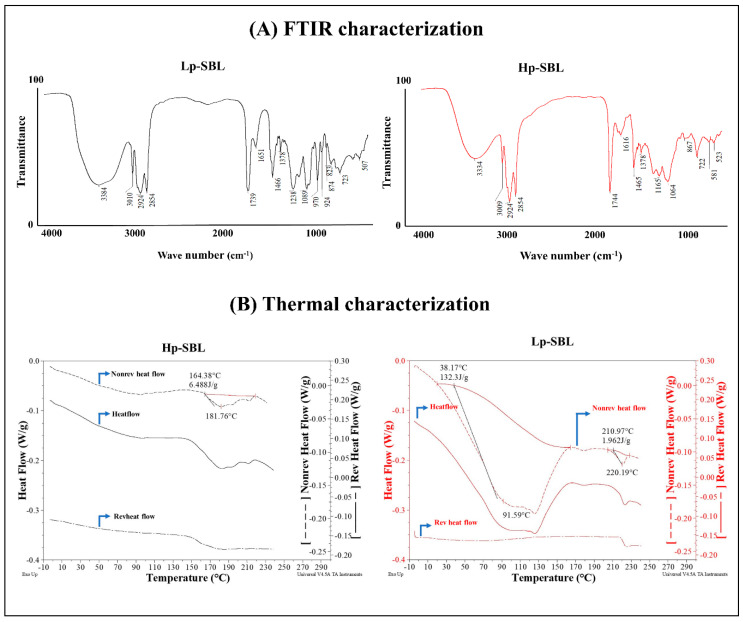
(**A**) Infrared spectroscopy (FTIR) signals for low purity grade (Lp-SBL) and high purity grade (Hp-SBL) soybean lecithins. (**B**) Modulated differential scanning calorimetry (mDSC) signals for low purity grade (Lp-SBL) and high purity grade (Hp-SBL) soybean lecithins. The solid line corresponds to the total heat flow, the short dashed line corresponds to the non-reversible heat flow, and the broken dashed line corresponds to the reversible heat flow.

**Figure 2 molecules-25-05344-f002:**
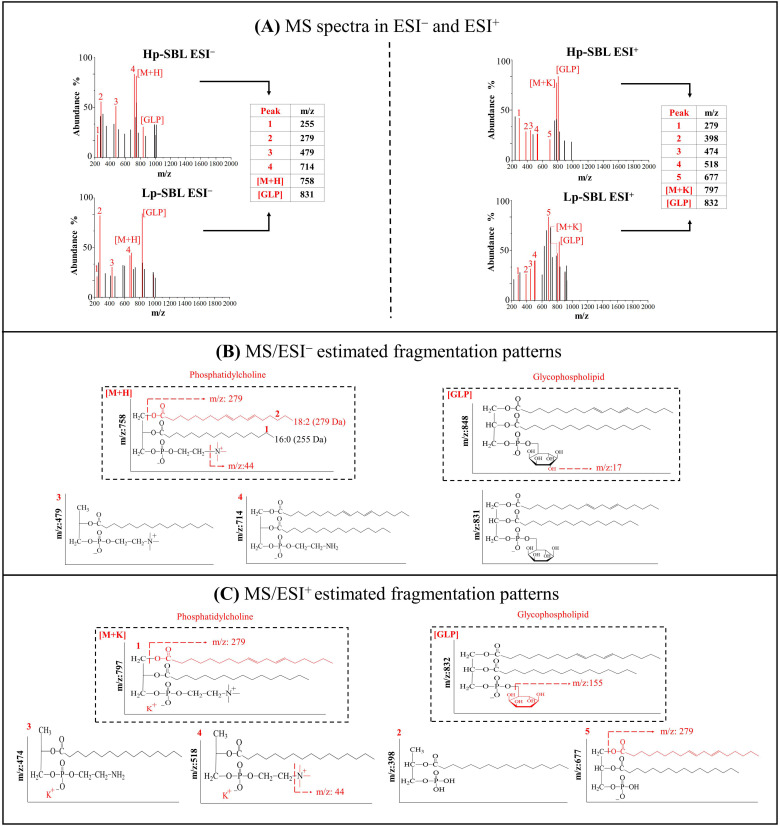
(**A**) Mass spectrometry (MS) spectra in total ion scanning mode for positive ion (ESI^−^) and negative ion (ESI^+^) modes for high purity lecithin (Hp-SBL) and low purity lecithin (Lp-SBL). (**B**) MS/ESI^−^ estimated fragmentation patterns. (**C**) MS/ESI^+^ estimated fragmentation patterns.

**Figure 3 molecules-25-05344-f003:**
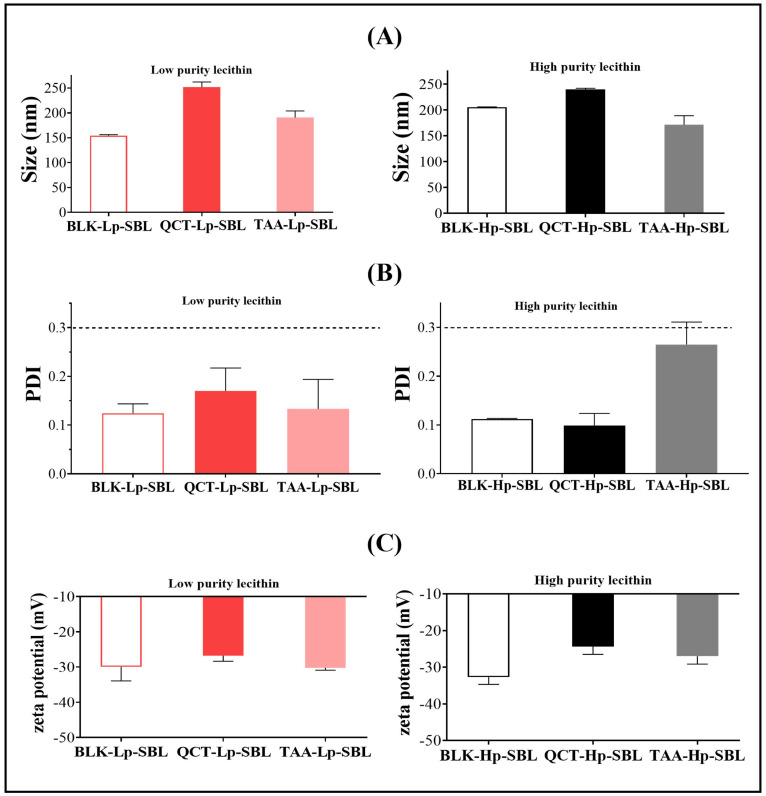
(**A**) Size, (**B**) polydispersity index (PDI), and (**C**) zeta potential for liposomes prepared with high purity lecithin (Hp-SBL) and low purity lecithin (Lp-SBL). BLK corresponds to unloaded nanoliposomes. Data are reported as the mean ± SD and n = 3.

**Figure 4 molecules-25-05344-f004:**
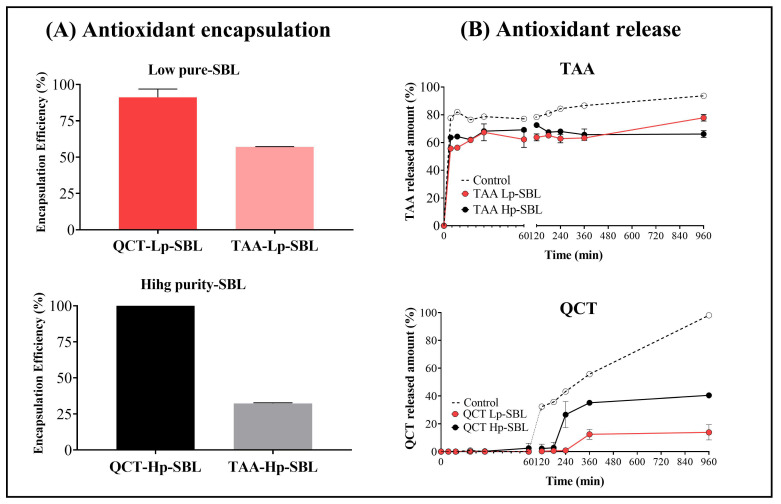
(**A**) Encapsulation efficiency and (**B**) release profiles of antioxidant (TAA and QCT) liposomes prepared with high purity lecithin (Hp-SBL) and low purity lecithin (Lp-SBL). QCT is quercetin and TAA is trans-aconitic acid. Data are reported as the mean ± SD and n = 3.

**Figure 5 molecules-25-05344-f005:**
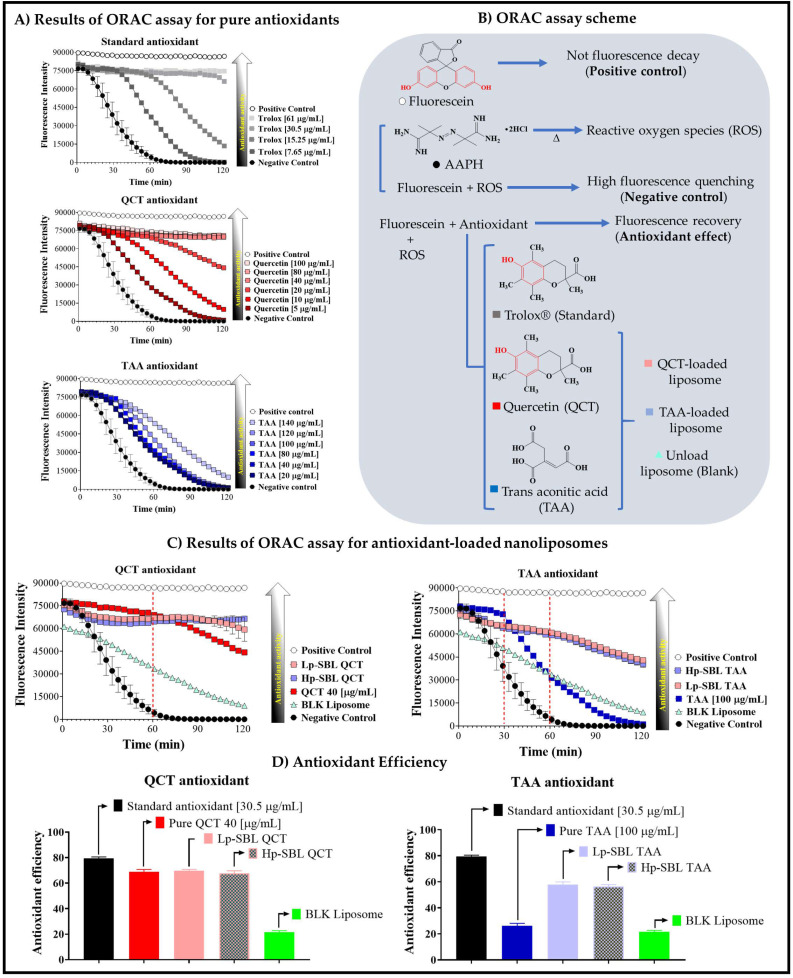
Results of the oxygen radical absorbance capacity (ORAC) assay. (**A**) Pure antioxidants. (**B**) ORAC method scheme. (**C**) Antioxidant-loaded nanoliposome. (**D**) Antioxidant efficiency. Data are reported as the mean ± SD and n = 3.

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
