# Peer review of "Development of Antioxidant-Loaded Nanoliposomes Employing Lecithins with Different Purity Grades"

_molecules, 2020, doi:10.3390/molecules25225344_

Round 1
Reviewer 1 Report
This manuscript is an interesting investigation of physicochemical properties and antioxidant capacities of antioxidant-loaded soybean lecithin nanoliposomes, with regards to how the polarity of antioxidants and the purity of soybean lecithin affect the results. This work is of interest because of the great attention paid by scientists and industries on how to improve the bioavailability and functionality of antioxidants in recent years.
This manuscript is well constructed, the methodologies are appropriate to the research objectives, and the results are clearly discussed. Notwithstanding, there is one arguable point - Although high and low purity soybean lecithins have similar FT-IR, MS, and DSC profiles, they are very different in chemical composition (e.g., 92% vs 50% phosphatidylcholine; lysophosphatidylcholine vs inositol phosphatides as the second predominant component), which represents another variable in addition to purity. Therefore my question is, how much of the difference in results is due to the difference in purity and how much is due to the difference in composition? The authors should give justifications for this.
In addition, there are some small modifications that could be done to improve the readability and quality of the article (texts and diagrams) and the preciseness of statements.
Line 29: "where quercetin showed a..." - quercetin → QCT (You have already defined the abbreviation and used it in the former part of this sentence, and you used TAA in the abbreviation in a parallel clause, so using QCT instead of quercetin comply with the consistency of expression.
Line 71-74: It seems that this part of texts has a different font style.
Line 147: The multiplication sign (×) in the equation looks like the letter "x".
Line 227: The numbers "42.7 °C", "98.92 °C" and "124.9 J/g" do not correspond with the ones on the Lp-SBL panel in Figure 1B (38.17 °C, 91.59 °C, and 132.3 J/g). Besides, the number "98.92 °C" has two decimal places while the other numbers reported in the same section have one decimal place.
Line 238: The number "164.8 °C" looks like a typo (on the graph it's 164.38 °C), while the numbers "181.7 °C" and "6.4 J/g" are not properly rounded (on the graph they are 181.76 and 6.488, which should be rounded to 181.8 and 6.5, respectively).
Line 248: the numbers "210.9 °C" and "220.1 °C" are not properly rounded (on the graph they are 210.97 and 220.19, which should be rounded to 211.0 and 220.2, respectively).
(Please let me know if there are specific rounding rules for DSC analysis.)
Line 252-253: The sentence stating which kind of line corresponds to which type of heat flow could be done as well-organized legends in the graph. I see that you put a short dash line with nonrev heat flow and a broken dash line with rev heat flow on the right y-axis of graphs but on the left y-axis where states heat flow (I assume this is the total heat flow), there is not a solid line.
Line 295: An extra period (.) is at the end of the sub-heading (inconsistent with other sub-headings).
Line 331, 349: It looks that you present the data as mean±SD, it would be good if you could add a sentence at the caption regarding how the data is reported (mean±SD) and how many replicate measurements you did for each treatment (n=).
Line 346: There are several typos in Figure 4A - "encapsultation" → "encapsulation"; "Low-pure" → "Low-purity"; "Hihg" → "High".
In Figure 4B, some of the data points have an error bar but some of them do not, is it because these at these data points you got identical values from replicate measurements, or because you only did one measurement?
Line 399: The "Figure 5C" in parenthesis should be in bold font, to be consistent with the format of other figure numbers when you refer in-text.
Author Response
the answers are in the attached file

Reviewer 2 Report
Dear Authors,
I read the manuscript "Development of Antioxidant-Loaded Nanoliposomes Employing Lecithins with Different Purity Grades” by C.H. Salamanca and coauthors.
The experiments and analyses are well conducted and properly explained and the results are interesting especially for food industry.
There are various references that are not complete. For instance article pages are not written in ref. 1, 2, 4, 8, 9,… 23, 24, 26 and so on. Please, check that all references follow editorial requirements.
I also found some phrases that are not clear, so I advise a general check of English language.
For instance:
line 101-012: Ethanolic solutions….were prepared from which volumes of 30…were taken. The phrase is very long: put a comma after “prepared” or start a new phrase such as “ from this solution volumes….were taken.
line 152: “For this” : at the beginning of the phrase it’s better “that’s why” or therefore.
line 379: “a similar trending” change with “a similar trend”
line 390-92: This phrase needs something (a verb?) or it is not clear: “a phenyl substituent can be appreciated and where a thermodynamic equilibrium between phenyl and phenolate species take place, being the phenolate form, the one that reacts with the ROS species”
line 441-442: This phrase is not clear or grammatically uncorrect: “it could be established that lecithins with a low purity degree and that consist by a greater quantity of polar phospholipids, these trend to encapsulate mainly trans-aconitic acid”
line 445: write “established” instead of “stablished”
line 458: “Maria J. Alhajj is mainly perform the test…” probably you want to say “Maria J. Alhajj mainly performed the test…” or “Maria J. Alhajj is mainly responsible for the test…”
For all these reasons, I believe this manuscript deserves to be published in Molecules after minor revision.
Author Response
the answers are in the attached file

Reviewer 3 Report
This was an excellent report on very thorough research. I have a few substantive questions/comments to consider:
- There was no mention of the limitations of the study. Also, mention how your results compare to another study, which was published very recently.
- The procedure of preparation nanoliposomes is not fully clear. For example, there was no temperature used, which can help with dissolving the mixture. How discard the solvents from the mixture. Liposome purification!!! how it was performed
- Please try to paraphrase the original paragraph of nanoliposomes preparation (check this published paper Decrease of Antimicrobial Resistance through Polyelectrolyte-Coated Nanoliposomes loaded with β-Lactam Drug)
- Abbreviations need to be defined for the first time.
- Poor resolution for all figures
- Space between numbers and units
- Conclusion needs to be revised, and pay more attention on innovations in this research
- Regarding the physicochemical characterization of nanoliposomes, it was preferable to do SEM or TEM in addition to measure particle size
- Academic writing should be objective. The language of academic writing should therefore be impersonal, and should not include personal pronouns, emotional language or informal speech. Use of personal pronouns (I/ my/ our/ us/ etc) can make the tone of writing too subjective
- New references are needed to show the recent works either in introduction section or discussion section
In conclusion, in my opinion, the work could be considered for publication in molecules Journal only after a throughout detailed revision of the complete manuscript.
Author Response
the answers are in the attached file

Reviewer 4 Report
Dear authors, I read your work with great interest. The manuscript is very well written and the topic is interesting. Below you can see the minor modifications that must be made to the manuscript. Good luck with your research in the future.
Line 67: change to "were purchased from Sigma-Aldrich"
Line 105: change to "(in the vortex) for 1 min and left to rest for 10 min"
Line 285: change to "Accordingly, it is important to highlight that the results suggest that the ions present in the low purity lecithin are also present in high purity lecithin"
Author Response
the answers are in the attached file

Round 2
Reviewer 1 Report
I am happy to see the revised version where many changes were made as per previous comments, however:
Figure 1B the legend for total heat flow (a solid line) is somewhat inconsistent with non-reversible heat flow (a short dash line in square brackets) and reversible heat flow (a broken dash line in square brackets).
Typos in Figure 4 that were pointed out in the first round of review were accepted according to their response letter, but have not been reflected in the document yet.
Minor revisions are required.
Regarding the question I raised in the first round of review, that both purity and composition of the two samples differed (not only purity) and if this have an effect on the results, the authors justified in the response letter. It would be nice if some justifications could be addressed in the article as a discussion of limitations of the current study and future research directions.
Author Response
Comment 1: Figure 1B the legend for total heat flow (a solid line) is somewhat inconsistent with non-reversible heat flow (a short dash line in square brackets) and reversible heat flow (a broken dash line in square brackets).
Answer: Figure 1B was modified, specifying what type of heat corresponds to each line in the DSC.
Comment 2: Typos in Figure 4 that were pointed out in the first round of review were accepted according to their response letter but have not been reflected in the document yet.
Answer: Indeed, it was. Therefore, we proceed to replace the figure correctly. We thank you again for such an observation
Comment 3: Regarding the question I raised in the first round of review, that both purity and composition of the two samples differed (not only purity) and if this have an effect on the results, the authors justified in the response letter. It would be nice if some justifications could be addressed in the article as a discussion of limitations of the current study and future research directions.
Answer: We consider to be more explicit regarding the sources from which the lecithins of this study come from and thus, include your very good comment in the manuscript.
Reviewer 3 Report
The manuscript was revised and re-written taking into consideration the reviewer’s suggestions and comments. The current version has been improved. However.it was preferable to do SEM or TEM because others do not provide an adequate particle size description
Author Response
The authors appreciate your comment.Nevertheless, we consider that the use of electron microscopy and specifically cryogenic microscopy, is also a good alternative for the determination of particle size. But it is not the only one or the best. In fact, it can be said that no technique is better than another for the determination of the particle size, but depends on the type of material or the type of system or another intrinsic condition of the sample (degradability, rapid aggregation, etc.). In the case of determining the particle size and polydispersity of liposomes in aqueous dispersion, the DLS technique is ideal. Beasides, this technique has been widely endorsed and reported for that purpose. On the other hand, if it were intended to know about the morphology of lipososms, microscopy techniques would be ideal. Therefore, the authors respectfully disagree with your comment that microscopic techniques are preferable to determine particle size.